# Applications of a working framework for the measurement of representative learning design in Australian football

**Peter R. Browne** [1,2]*, **Carl T. Woods**[1], **Alice J. Sweeting**[1,2], **Sam Robertson**[1,2]

**1** Institute for Health and Sport (iHeS), Victoria University, Footscray, Victoria, Australia, **2** Western Bulldogs, Footscray, Victoria

* peter.browne2@live.vu.edu.au

**Data Availability Statement:** All relevant data to reproduce the study are available within the manuscript and its Supporting Information files.

## Abstract

Representative learning design proposes that a training task should represent informational constraints present within a competitive environment. To assess the level of representativeness of a training task, the frequency and interaction of constraints should be measured. This study compared constraint interactions and their frequencies in training (match simulations and small sided games) with competition environments in elite Australian football. The extent to which constraints influenced kick and handball effectiveness between competition matches, match simulations and small sided games was determined. The constraints of pressure and time in possession were assessed, alongside disposal effectiveness, through an association rule algorithm. These rules were then expanded to determine whether a disposal was influenced by the preceding disposal. Disposal type differed between training and competition environments, with match simulations yielding greater representativeness compared to small sided games. The subsequent disposal was generally more effective in small sided games compared to the match simulations and competition matches. These findings offer insight into the measurement of representative learning designs through the non-linear modelling of constraint interactions. The analytical techniques utilised may assist other practitioners with the design and monitoring of training tasks intended to facilitate skill transfer from preparation to competition.

## Introduction

A predominant challenge facing sports practitioners is the design and implementation of training environments that represent competition. This approach to training design has been referred to as representative learning design (RLD) [1]. Theoretically, RLD advocates for training to consist of key (informational) constraints that are experienced within competition to maximise the transfer of skill from training to competition [1, 2]. Constraints are categorised into Individual (e.g., physical attributes and emotions), Task (e.g., rules and ground dimensions) and Environmental (e.g., weather and gravity) classes [3, 4]. To assist with the design of representative training tasks, practitioners typically record the constraints of a competitive environment to ensure such constraints are designed into training [5]. However,

**Funding:** The authors PRB, AJS and SR are affiliated with the Western Bulldogs Football Club. The funder provided support in the form of salaries for AJS and SR, and a volunteer position for PRB, but did not have any additional role in study design, data collection and analysis, decision to publish, or preparation of the manuscript. The specific roles of these authors are articulated in the 'author contributions' section.

**Competing interests:** The authors have read the journal's policy and have the following competing interests: AJS and SR are paid employees of the Western Bulldogs Football Club. PRB is a volunteer at the Western Bulldogs Football Club. This does not alter our adherence to PLOS ONE policies on sharing data and materials. There are no patents, products in development or marketed products associated with this research to declare. CTW has no competing interests to declare.

understanding how these constraints interact to influence a performer's actions and behaviours is an ongoing challenge for practitioners given the non-linearity and dynamicity of sports performance [6].

An important feature of a constraints-led approach to training design is the understanding that constraints do not exist in isolation. Rather, they dynamically interact with one another, often in a continuous manner [4, 7]. However, the measurement of the dynamic interaction of constraints has been somewhat neglected within the literature [6]. Whilst constraints can be collected from training and competition environments, such approaches often overlook constraint interaction and are unable to capture then analyse the complexity of systems in full [8, 9]. Recently, the interaction among constraints was examined via machine learning techniques in Australian football (AF) [6, 10]. The application of a rules-based approach enables the complexity of RLD to be measured, through the identification of key constraint interactions based on both their frequency and their displayed influence on behaviours. An informed RLD is vital for practitioners, as how constraints are enacted in training implicates skill development and learning transfer [1, 11–13].

Within many team sports, including AF, small sided games (SSGs) are used as a frequent training modality due to their perceived representativeness of competition matches and ease of constraint manipulation [14, 15]. Specifically, SSGs can be used to simulate sub-phases of competition, whilst to some extent, preserve the complex interactions between an athlete and their environment [16–18]. Match simulations are another common training strategy within preparation for performance models in team sports, as they afford practitioners with a practice landscape that can simulate scenarios commonly encountered within competition. Match simulations and SSGs are different types of training modalities and thus, the frequency and interaction of constraints may differ. The intent of these training modalities are different, and as such, their use within the broader training schedule should be carefully considered by coaches [19].

The primary aim of this study was to compare constraint interactions and their frequencies, between match simulations, SSGs, and competition matches in AF. These comparisons were facilitated using a rule-based algorithm. Secondly, the study aimed to determine the extent to which they influenced disposal type and effectiveness. Thirdly, this study sought to understand the sequential nature of disposals by examining whether the efficiency of a disposal was influenced by the preceding disposal. By addressing these aims, this study sought to progress the methodology of measurement for RLD in sporting environments.

## Methodology

Data were collected from official matches and training sessions from one Australian Football League (AFL) club across the 2018 and 2019 (pre)seasons. All 2018 regular season matches were included ($n = 22$, disposal instances = 3,478). Specific tasks from training sessions were included, consisting of match simulations ($n = 13$, disposal instances = 1,298) and SSGs ($n = 24$, disposal instances = 2,677). Seven versions of SSGs were included ranging from seven to 18 athletes. Ground dimensions ranged from approximately 20 x 20 m to 60 x 60 m. Number inequalities were included in some SSGs, with the largest discrepancy between team numbers being three additional attackers compared to defenders. Given the applied nature of this research, these design features were hard to control. Ethical approval was granted by the University Human Research Ethics Committee (application number: HRE18-022), and written consent was gained from the organisation to use de-identified data as collected per regulation training practices.

**Table 1. Description of constraints sampled, their sub-category, and definition.**

| Constraint sampled | Sub-category | Definition |
|---|---|---|
| Disposal Type | Kick | Disposal of the football with any part of the leg below the knee |
| | Handball | Disposal of the football by hitting it with the clenched fist of one hand, while holding the football with the other |
| Pressure | Pressure | Opposition player defending the ball carrier from any direction |
| | No Pressure | |
| Time in Possession | > 2 sec | Time with ball in possession from receiving the football to disposing of it |
| | < 2 sec | |
| Disposal Effectiveness | Effective | An effective kick is of more than 40 m to a 50/50 contest or better for the team in possession, or a kick of less than 40 m that results in retained possession |
| | Ineffective | |

Match footage was provided by Champion Data (Melbourne, Australia, Pty. Ltd.), whilst training tasks were filmed by club staff from the same perspective as the competition match footage (behind the goals and side view). All footage was then subjected to notational analysis via SportsCode (version 11.2.3, Hudl). The lead author and a performance analysts coded all footage using a code window developed with a weighted kappa statistic of >.80, indicating very good reliability [20]. Constraints collected included: disposal type, pressure, time in possession and disposal effectiveness (Table 1). These constraint types were based upon similar literature [6, 10, 21]. The nature of the options for each constraint sampled limited bias in the rule-based approach, as all constraints had the same number of sub-categories (Table 1).

All analyses were undertaken in the R computing environment (version 3.6.1, Vienna, Austria) and included a three-stage process. All code for the following analyses are available on Github (www.github.com/PeterRBrowne). First, association rules were generated for all disposals for match simulations, SSGs and competitive matches. Association rules are a type of machine learning algorithm which can identify underlying and frequent non-linear patterns in a large dataset [22]. The 'Arules' package was used to apply the Apriori algorithm [23] and to measure the association between multiple constraints on disposal efficiency. Minimum support and confidence levels were set at 0.0002 to allow for all possible rules to be generated. The minimum number of variables was set at four to ensure that each coded constraint was included. The association rules were arbitrarily assigned an alphabetical identity (ID), being then compared by levels of support and confidence [24].

The frequency with which a rule occurred and was then followed by a subsequent rule was then calculated using the 'tidyr' and 'dplyr' packages [25, 26]. The difference between training and competition frequencies was then calculated. The observed frequency of a third disposal being effective was calculated. This was visualised using a lattice plot, with colour hues to differentiate the observed frequency of an effective disposal. The level of observed frequency of an effective disposal was calculated as the weighted average of the confidence of a Rule ID and the frequency with which three sequential rules occurred.

## Results

The association rules with assigned alphabetical ID are presented in Table 2, and the differences in rule frequency (A) and confidence levels (B) are displayed in Fig 1. The lowest support across all three environments was Rule E (0.012), and the largest was Rule G (0.316), with both occurring in the competition environment (Fig 1A). The support levels for match simulation rules were generally more representative of a competitive match, relative to SSGs, based on the constraints measured. Rule G, a pressured handball performed within 2s, showed the largest

**Table 2. Breakdown of each possible association rule and its associated alphabetical ID.**

| ID | Type | Pressure | Time in Possession (seconds) |
|----|------|----------|------------------------------|
| A | Kick | No Pressure | <2 |
| B | Kick | No Pressure | >2 |
| C | Kick | Pressure | >2 |
| D | Kick | Pressure | <2 |
| E | Handball | No Pressure | >2 |
| F | Handball | No Pressure | <2 |
| G | Handball | Pressure | <2 |
| H | Handball | Pressure | >2 |

difference between competition matches and the SSGs (Fig 1A). Levels of support also varied between environments, with Rule G being the most frequent in matches and match simulations, whilst Rule D was the most frequent in SSGs (Fig 1A). With the exception of Rule C, rules corresponding to 'kicks' yielded lower confidence in competition matches relative to SSGs, but higher confidence relative to match simulations (Fig 1B). For rules relating to 'handballs', the confidence was highest in competition matches relative to the training tasks (Fig 1B).

The differences between sequential Rule IDs were calculated between training and competition environments (Table 3). Positive values reflect a greater frequency of occurrence within competition matches, whereas negative values indicate greater frequency of occurrence in the training environment. Match simulations were more similar to competition matches, relative to SSGs in levels of support (Table 3A) and confidence (Table 3B). Disposal sequence differed more between competition matches and SSGs, with eight sequences having a greater than a ±20% difference between environments (Table 3B). Whilst for similar disposal sequences between training and competition environments, both match simulation and SSGs were similar with twelve and eleven sequences having less than ±1% difference respectively (Table 3).

Fig 2 depicts the observed frequency of effectiveness of the third disposal following two sequential disposals across competition matches (A), match simulations (B) and SSGs (C). The variation between competition and training environments are visualised through colour hues, in addition to the observed frequency being overlayed. The third disposal in the sequence was more likely to be effective in SGGs, relative to competition matches and match simulation. Specifically, the observed frequency of the third disposal in the sequence being effective ranged from 54 to 89% for competition matches, 49 to 84% for match simulations, compared to 77 to 88% for SSGs (Fig 3). The majority of competition match third disposal effectiveness was above 70%, with only six disposal sequences less than 70% effectiveness. Comparatively, 28 disposal sequences during match simulations resulted in less than 70% effectiveness (Figs 2 and 3).

## Discussion

The primary aim of this study was to compare constraint interactions and their frequencies between match simulations, SSGs, and competition matches in AF using a rule-based algorithm. Secondly, it aimed to determine the extent to which they influenced disposal type and effectiveness. Thirdly, this study sought to understand the sequential nature of disposals and whether disposal sequences are dependent upon the preceding disposals. Accordingly, this study aimed to progress the methodology for the measurement of RLD beyond recording a

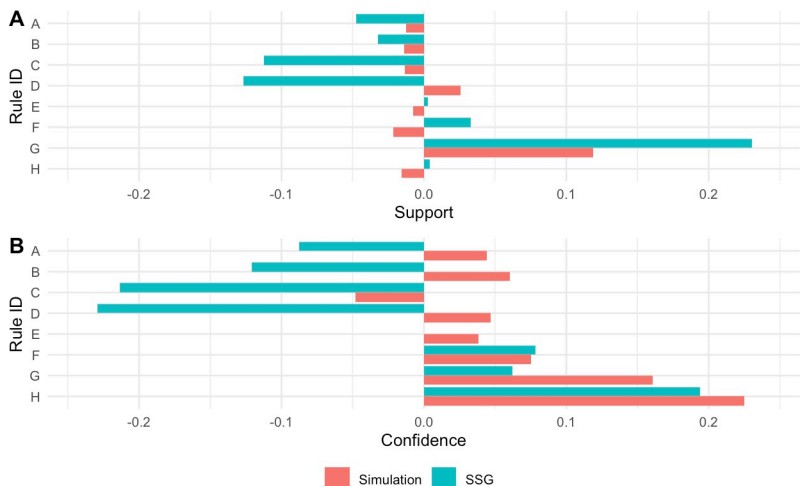

**Fig 1.** Variation in levels of support (A) and confidence (B) of each Rule ID match simulations and Small Sided Games relative to competition matches. Where zero is equal to competition matches. Positive values reflect greater values for competition matches.

single instance and to account for continuous nature of match events and the constraints impacting them. For instance, by extending the rule-based approach from exploring a single disposal as in Browne, Sweeting [10], this study sought to understand disposal sequences, and

**Table 3. Difference between frequency of second pass following first pass for competition matches and match simulations (A), and competition matches and SSGs (B).** Values are expressed as percentage differences (%).

**A**

| | | Second Pass | | | | | | | |
|---|---|---|---|---|---|---|---|---|---|
| | | A | B | C | D | E | F | G | H |
| First Pass | A | -2 | 9 | 3 | 2 | 1 | -6 | -6 | -2 |
| | B | 4 | -18 | 2 | 9 | -3 | -3 | 11 | -2 |
| | C | 0 | 7 | -9 | 5 | 0 | -2 | 1 | -2 |
| | D | -1 | 5 | -11 | 2 | 0 | -4 | 12 | -4 |
| | E | 3 | -18 | 13 | 0 | NA | 8 | 0 | -5 |
| | F | -2 | -7 | 0 | 6 | 0 | 3 | 3 | -3 |
| | G | -7 | 1 | 2 | -1 | -1 | -4 | 11 | -2 |
| | H | -8 | -3 | -2 | 0 | 0 | -2 | 23 | NA |

**B**

| | | Second Pass | | | | | | | |
|---|---|---|---|---|---|---|---|---|---|
| | | A | B | C | D | E | F | G | H |
| First Pass | A | -18 | 4 | -4 | -7 | 3 | 6 | 16 | 1 |
| | B | -4 | -24 | 0 | 6 | 0 | 0 | 23 | -1 |
| | C | -3 | 19 | -22 | -26 | 0 | 3 | 28 | 2 |
| | D | -1 | 13 | -20 | -26 | 0 | 4 | 28 | 1 |
| | E | 2 | -8 | -4 | 2 | NA | 8 | 2 | -2 |
| | F | -7 | -2 | -1 | 0 | -1 | -2 | 12 | 0 |
| | G | -2 | 4 | -2 | 3 | 0 | -3 | 2 | -3 |
| | H | -3 | -3 | -3 | -19 | 0 | 5 | 26 | NA |

*Note*: Greater negative values (the deeper the orange hue) indicate greater frequency of the rule sequence in the training environment. Larger positive values (the deeper the blue hue) indicate a greater frequency of the rule sequence in the competition environment. *NA* represents where the two rule IDs did not occur sequentially. Values closer to '0' denote closer similarities between training and competition.

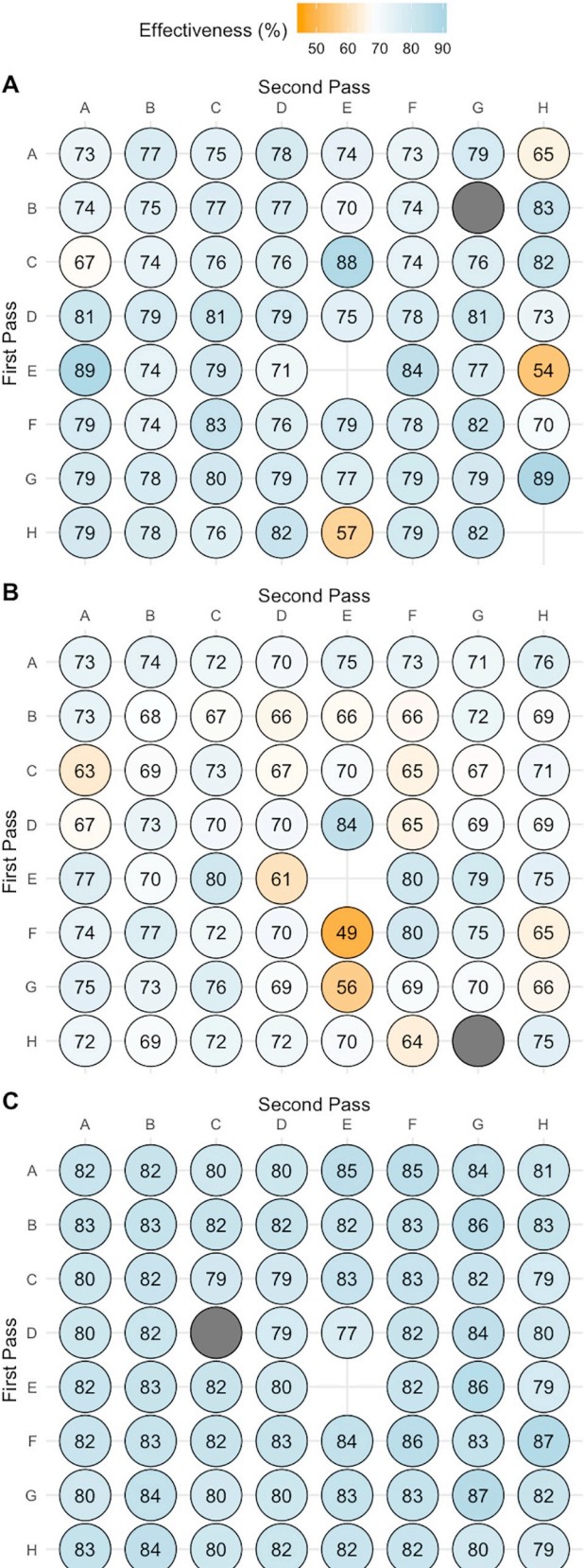

**Fig 2.** The observed frequency of effectiveness of the third disposal following two sequential disposals across competition matches A) competition match B) match simulation C) Small Sided Games. Values expressed as percentages (%). *Note*: The scale moves from orange to blue with the deeper the hue the greater observed frequency of an effective third disposal. Blank sections are those which did not have two sequential passes. Grey circles reflect those sequences of passes which did not continue to a third disposal.

the extent to which disposals are dependent on preceding disposals. Results demonstrated that the frequency and confidence of different disposal types and constraint interactions varied between match and training environments. These differences varied depending on the training task, with match simulations yielding a greater level of representativeness to matches relative to SSGs. However, the level of representativeness and intent of each SSG may differ, the efficacy of each approach may vary depending on the given context.

With respect to the primary aim, this study demonstrates that an understanding of the differences between support and confidence levels of constraint interactions within training and competition environments is an important consideration for the design of representative training tasks. For example, match simulations generally showed greater similarity to competition matches, with respect to disposal type. However, competition matches incurred a greater frequency of pressured handballs performed within 2s (Rule G), relative to match simulations. These differences between training and competition environments could exist for a number of practical reasons. Notably, the design features of the SSGs could intentionally favour a specific disposal type (e.g. kick), whilst, in general, the training environment could incur less physical pressure relative to a competitive match, given differences in physical exertion and intensity [21]. Practitioners may therefore use this information to better understand the influence of constraints on performance, which could, improve the representativeness of such training tasks through informed manipulation, such as increasing (or decreasing) playing field dimensions to encourage differing levels of pressure on disposals [27, 28]. Moreover, this study provides a methodology to better understand the design of training tasks that may aid practitioner decision-making in implementing appropriate SSGs for their desired intent.

Due to the intent of the match simulations compared to SSGs, it is unsurprising to note that match simulations were more representative of competition compared to the SSGs. Thus, it is reasonable to suggest that not all training tasks will yield the same level of representativeness,

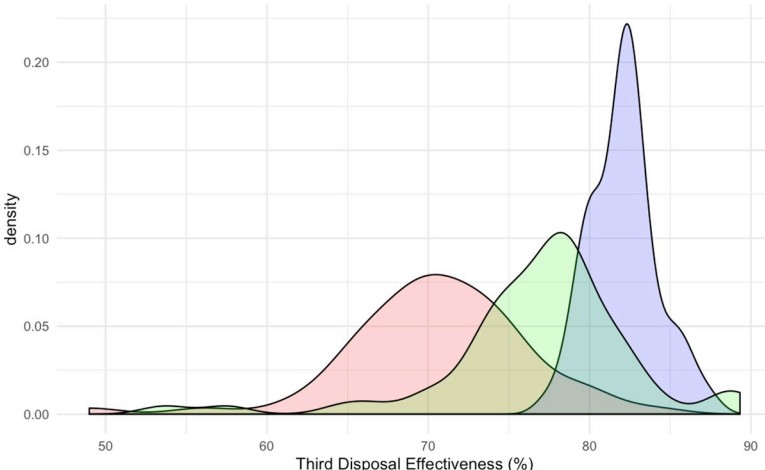

**Fig 3. Density plot of the observed frequency of effectiveness of a third disposal.** This was based upon the previous two disposals across competition matches (green), match simulations (red) and Small Sided Games (blue).

potentially due to their explicit intent. For example, a practitioner may manipulate certain constraints of a SSG to facilitate greater disposal efficiency, reducing representativeness relative to competition, but still achieving the intended task goal. Conversely, a practitioner may want to challenge disposal efficiency within a SSG by manipulating temporal and spatial constraints so the training task is harder (with reference to time and space) than what is afforded within competition. Although it is likely that practitioners' do not plan for every training task to express near perfect representativeness, this methodology provides a platform by which 'target' areas could be identified, informing practitioners as to how frequent non-representative actions are performed within practice. Such information could better guide the macro- and meso- structures of practice, ensuring less representative training tasks are coupled with more representative tasks. Further, this aligns with the principles of periodisation for skill acquisition [19], emphasising the importance of being able to measure the influence of constraint interaction within training tasks [10]. A training task classification systems may be able to aid practitioners in this process to ensure the appropriate tasks are conducted together based on its characteristics and intent [29]. However, the ideal balance of representative versus non-representative practice to gain the greatest performance benefit in competition, is currently unknown.

The third aim sought to explore concomitant disposal sequences. Differences between the training and competitive performance environments were found when exploring the observed frequency of a third sequential disposal being classified as 'effective'. Understanding disposal sequences is a key feature of complexity. This is essential for RLD as it enables understanding of not just the current status, but which interactions occur after. For matches and match simulations, the observed frequency of an effective third sequential disposal was lower compared to the SSGs. This practice task yielded the highest range of observed frequency for an effective third disposal, likely due to the task design of the SSGs, which may encourage a more continuous, effective, chain of disposal. This could have been intentionally designed within the SSGs through the systematic manipulation of player numbers (task constraint) to favour the offensive team (for example, a SSG consisting of 6 vs. 4). Nonetheless, this analysis demonstrates how a chain of disposals could partially shape future disposal effectiveness, thus providing some evidence that the effectiveness of a disposal may not be independent from preceding events.

A limitation of this study was that it grouped all SSGs together, despite it being possible that some SSGs had differing task intentions and subsequent challenge points, diluting their representativeness. Accounting for intent in SSGs may allow for a more complete insight into their representativeness relative to competitive matches. Future research can look to apply the methodological advancements from this study to further understand the differences between various SSGs. Additionally, a limited number of constraints were used to model RLD, and thus the model presented here is a truncated view of RLD. The sampling of appropriate constraints is an evolving process, as better and new measures become available. The use of experiential coach knowledge could aid in the informed selection of constraints, however experiential knowledge is dependent on the individual, subject to biases and the environment in which it is applied. Further, this study focused solely on the ball carrier, with it being likely that other constraints, such as opposition and teammate location and the individual's action capabilities, additionally influenced the disposal outcome. Models that consider these factors will likely further explain disposal effectiveness, but their performance must be considered against any decrease in interpretability that may arise from the utility of larger constraint sets. Additionally, future studies could look to examine the frequency of rule occurrence in a defined period of time [21]. For instance, a SSG played in a small area may have a higher frequency of disposals per minute, compared to a larger area.

## Conclusion

Disposals are influenced by the interaction of constraints in training and competition environments in elite AF. Variation exists in the frequency whereby disposals occur under specific constraints across the competition matches, match simulations and SSGs. Although training and competition environments differed, this study found greater levels of representativeness existed between match simulations and competition matches compared to the grouped SSGs and competition matches. These insights can aid the comprehension of how constraints interact to shape the emergence of specific disposals and their effectiveness, affording practitioners with a platform for the development and measurement of representative training tasks. The analytical techniques applied in the present study are not limited to AF and may assist in designing representative training tasks across other sports via the consideration of constraint interaction. Importantly, this study provides a methodological advancement in the measurement of constraint influence, frequency and accounting for the continuous nature of sport.

## Supporting information

**S1 Data.**
(CSV)

## Author Contributions

**Conceptualization:** Peter R. Browne, Carl T. Woods, Alice J. Sweeting, Sam Robertson.

**Data curation:** Peter R. Browne.

**Formal analysis:** Peter R. Browne.

**Methodology:** Peter R. Browne, Alice J. Sweeting, Sam Robertson.

**Supervision:** Carl T. Woods, Alice J. Sweeting, Sam Robertson.

**Visualization:** Peter R. Browne, Carl T. Woods.

**Writing – original draft:** Peter R. Browne.

**Writing – review & editing:** Peter R. Browne, Carl T. Woods, Alice J. Sweeting, Sam Robertson.

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
