## [Decision Letter · Decision Letter 0]

11 Sep 2020

PONE-D-20-14878

Applications of a working framework for the measurement of representative learning design in Australian football

PLOS ONE

Dear Mr. Browne,

Thank you for submitting your manuscript to PLOS ONE. After careful consideration, we feel that it has merit but does not fully meet PLOS ONE’s publication criteria as it currently stands. Therefore, we invite you to submit a revised version of the manuscript that addresses the points raised during the review process.

We look forward to receiving your revised manuscript.

Kind regards,

Slavko Rogan

Academic Editor

PLOS ONE

Journal Requirements:

3. Thank you for including your competing interests statement; "The authors have declared that no competing interests exist."

We note that one or more authors are affiliated with;  Western Bulldogs, Footscray.

Reviewers' comments:

Reviewer's Responses to Questions

**Comments to the Author**

1. Is the manuscript technically sound, and do the data support the conclusions?

Reviewer #1: Yes

Reviewer #2: Yes

2. Has the statistical analysis been performed appropriately and rigorously? 

Reviewer #1: Yes

Reviewer #2: Yes

3. Have the authors made all data underlying the findings in their manuscript fully available?

Reviewer #1: Yes

Reviewer #2: Yes

4. Is the manuscript presented in an intelligible fashion and written in standard English?

Reviewer #1: Yes

Reviewer #2: Yes

5. Review Comments to the Author

Reviewer #1: In general, the article presents a problem of great relevance, is well written and presents methodological advances related to the analysis of the degree of representativeness of training tasks. However, some corrections are required and all comments were included in the attached file.

Reviewer #2: The manuscript “Applications of a working framework for the measurement of representative learning design in Australian football” compared constraint interactions and their frequencies in training (match simulations and small sided games) with competition environments in elite Australian football. The paper brings new and important knowledge for practitioners and I recommend the manuscript publication. The comments provided below aimed to improve the quality of the manuscript.

• Methods

Reliability issues - According to the authors, “All footage was then subjected to notational analysis via SportsCode (version 11.2.3, Hudl).”

It is important to provide more information about the notational analysis to assure its reliability. Was the analysis performed by one or two authors? Do they have experience in codification? Is there any measure of reliability?

• Results

The authors performed a descriptive analysis of tables A and B. Is it possible to perform an inferential analysis to corroborate the descriptive findings?

• Discussion

I agree with the authors that it is unsurprising to note that match simulations were more representative of competition compared to the SSGs. Are there studies that show similar results in other sports? If so, which methods were used by the previous studies? This discussion may highlight the advantages of the currently proposal.

• Limitation

The SSGs were analyzed together, but it was already recognized by the authors.

6. PLOS authors have the option to publish the peer review history of their article (what does this mean?). If published, this will include your full peer review and any attached files.

Reviewer #1: No

Reviewer #2: **Yes: **Júlia Barreira

---

## [Author Response · Author response to Decision Letter 0]

4 Oct 2020

Reviewer 1:

Abstract:

Line 31: What do the authors mean by "disposal type and effectiveness across environments"?

• Response: This sentence has been amended to improve its clarity – please refer to line 7-9

Line 32/33: What constraints? What do the authors mean by association rules?

• Response: Constraints are defined here as boundaries in which skills emerge and shape their outcome, such as opposition pressure. Association rules are a machine learning approach that analyses the underlying and interacting patterns in data, helping us to understand non-linear interactions in greater detail. 

• We have since amended this sentence to better define both constraints and association rules (please see line 9-11). Note, however, word restrictions on the abstract do limit our capability to expand on these further.

Line 34: Difficult to follow

• Response: This sentence has been amended to improve its clarity – please refer to line 11-13.

Introduction:

Line 52: Something to reflect on: Is that really the biggest challenge? Does the coach want to reproduce the same constraints present in the competitive environment? Is that the intention? If so, there is nothing more representative than match simulations and the competition environment itself. Should we want to reproduce that same environment in training contexts? Or should we develop players' competencies and skills through tasks that maintain key sources of information that guide their actions and decisions? How to prepare youth players to deal with different types of constraints through training tasks with less difficulty and complexity than competitive environment and simulated matches? Here's the big challenge!

• Response: This is an interesting point raised by the reviewer. Indeed, the goal of training designs is not to replicate competition, but is to ensure that key sources of information experienced in competition are present in training activities so that the relevant information-movement couplings are preserved. As correctly alluded to by the reviewer, this may not always ‘look’ like a game. Further, it is why we refer to representative learning design being the process of maintaining the key information constraints within the first paragraph (Line 25 – 28) – the role of the coach is to therefore manipulate these informational constraints to make training easier (or harder) to promote athlete learning. Also, there is a lot that we still don’t know about how constraints influence performance in competition and training environments.

Line 76/77: Here we need to be very careful. Can both SSCG and Match simulations be considered training tasks? I say this, because in fact the match simulations will present a high level of representativeness, as well as larger SSCG configurations. But, because of that, will I always use match simulations during training sessions? Or do we need to think about development through different stages, preparing athletes to manage the constraints of simulated matches and, consequently, with the competition? Otherwise, I can lead readers to believe that it would be better to just train through simulated matches.

• Response: This is again an interesting point raised by the reviewer. Firstly, it is our perspective that both are training tasks, as both are performed in a practice environment with the primary intent of improving multiple aspects of performance in competition. Secondly, it was not our intent to suggest that training should only consist of these modalities, and indeed, their use should be matched to the broader practice schedule used by the coach or organisation. Please refer to lines 56-58 where we have clarified this.

Line 81/82: Difficult to follow

• Response: This has been amended accordingly – please refer to line 63-65.

Methods: 

Line 89: We need more information about those SSCG used, regarding players number, pitch dimension, rules, etc. These constraints will affect players performance during SSCG. 

• Response: Given the reviewers comment, we have included additional information regarding the general design features of the SSGs used. Please refer to lines 73-77 in our revised manuscript. We also acknowledge the reviewer’s consideration of our cited limitation of the SSGs stated in our original submission. 

Line 101/Table 1: Here is my biggest doubt: to measure representativeness, can we use only ball carrier actions? Technical skills? What about actions without a ball?

• Response: This is a fair point raised, and clearly, this additional information in practice will be crucial in better understanding the representativeness of task designs. Given the proposal of a new methodological way of analysing data (not necessarily capturing it), we were unable to include a greater array of constraints impacting upon representativeness. Accordingly, we have included an additional section of text describing this as a study limitation – please refer to lines 254-256.

Discussion: 

Line 192: Again, I agree that the simulated matches always present this highest level of representativeness. But, could these results have been influenced by the SSCG used in the study? We don't have any information about the types of games used and which key tasks constraints were manipulated, and for me that makes all the difference!

• Response: We now feel that this issue should be partly alleviated for the reviewer in amendments made in response to their earlier comments (please see lines 73-77 for additions on the SSG design features). Further, we absolutely agree with the reviewer that grouping the design features of the SSGs into one training modality could have somewhat diluted their representativeness, and is why we specifically asserted this as a limitation to our work in our original submission from line 247-256.

Line 205: Exact! For me, the biggest problem is in the design process of training tasks, and not in using SSCG or not. The big challenge is to know which SSCG to use, when to use it and for whom to use it.

• Response: We are glad the reviewer agrees with our sentiments here, and wish to further clarify them in our response. Specifically, we do not wish to propose that SSGs or match simulations are the only training modalities that should be used by coaches in high performance sport. As suggested by the reviewer a few times, the challenge for coaches is not to just use these modalities, but to understand when and why such modalities would be used in the broader training schedule.

• To help clarify these contentions, please refer to lines 204-213 where we indicate that our results could help coach decision-making in implementing SSGs for their desired intent.

Conclusion:

Line 265-267: I suggest that the authors be careful with the way they put that statement. In fact, the authors found this, but we need to remember the limitations mentioned above, mainly of not having considered the different constraints manipulated in SSCGs, which will have impacted the results found for the SSCG.

• Response: We thank the reviewer for highlighting this, it is a fair point, and we have since softened our wording in light of the reviewer’s suggestion (lines 277-279). Specifically, we have encouraged future research to extend upon our findings by applying the analytical approaches we used to compare differing SSGs (as opposed to grouping like was done in our study) and their level of representativeness to competition matches (please see lines 254 - 256).

Line 272: “Importantly, this study provides a methodological advancement in the measurement of constraint influence, frequency and accounting for the continuous nature of sport” In my opinion, this should be the focus of this study. The authors seem to value comparisons between contexts (competitive and training) more than in methodological advances to assess representativeness.

• Response: We feel that the paper does indeed centralise the progression of how representative design is measured in high performance sport – it is stated in the title of our paper. Furthermore, we cloak our aims within a statement that actually asserts the main objective of this study is to progress the measurement of representative learning design on line 66 and we have specifically referred to this advancement in the first paragraph of our discussion from line 183 – 186. Of course, to show the use of our methodology, we had to compare training modalities, and it would have been remiss to not appropriately discuss these modalities and subsequent results throughout the paper.

Reviewer 2:

Methods: Reliability issues - According to the authors, “All footage was then subjected to notational analysis via SportsCode (version 11.2.3, Hudl).”

It is important to provide more information about the notational analysis to assure its reliability. Was the analysis performed by one or two authors? Do they have experience in codification? Is there any measure of reliability?

• Response: The data was collected by the lead author and a professional performance analyst. Both had multiple year experience in manual coding of team sport data. Furthermore, the methods and code windows used have previously been tested for reliability and were found to have a weighted kappa statistic of >0.80 (Back, 2015). Accordingly, we have included additional information in our revised manuscript that answers the questions raised here by the reviewer. Please refer to lines 82-84.

Results: The authors performed a descriptive analysis of tables A and B. Is it possible to perform an inferential analysis to corroborate the descriptive findings?

• Response: We thank the reviewer for this comment. The intent was to use a rule-based approach to compare training and competition environments, and as such, including an inferential analysis here, which does not specifically answer an aim of the study, may unnecessarily confuse readers. Therefore, in this instance, we have decided against making an amendment to our manuscript.

Discussion: I agree with the authors that it is unsurprising to note that match simulations were more representative of competition compared to the SSGs. Are there studies that show similar results in other sports? If so, which methods were used by the previous studies? This discussion may highlight the advantages of the currently proposal.

• Response: Unfortunately, to the best of our knowledge, there are no studies which have compared match simulations and SSGs with competition matches, with respect to technical actions and the constraints captured. Indeed, studies have compared training activities with competition environments (Ireland et al., 2019) and others have looked at how manipulating SSGs influences physiological and tactical outputs (for example, Klusemann et al, 2012, Goncalves et al., 2017). However, these studies explore tactical components of performance and/or use discrete frequency counts of events, which is where our work has looked to advance, by using a rule-based approach. Please refer to lines 183-189 where we have highlighted this intent more directly. 

Limitation: The SSGs were analysed together, but it was already recognized by the authors.

• Response: We acknowledge the reviewer’s consideration of our cited limitation of the SSGs stated in our original submission.

---

## [Decision Letter · Decision Letter 1]

2 Nov 2020

Applications of a working framework for the measurement of representative learning design in Australian football

PONE-D-20-14878R1

Dear Mr. Browne,

We’re pleased to inform you that your manuscript has been judged scientifically suitable for publication and will be formally accepted for publication once it meets all outstanding technical requirements.

Kind regards,

Slavko Rogan

Academic Editor

PLOS ONE

Additional Editor Comments (optional):

Reviewers' comments:

Reviewer's Responses to Questions

**Comments to the Author**

1. If the authors have adequately addressed your comments raised in a previous round of review and you feel that this manuscript is now acceptable for publication, you may indicate that here to bypass the “Comments to the Author” section, enter your conflict of interest statement in the “Confidential to Editor” section, and submit your "Accept" recommendation.

Reviewer #1: (No Response)

Reviewer #2: All comments have been addressed

2. Is the manuscript technically sound, and do the data support the conclusions?

Reviewer #1: (No Response)

Reviewer #2: Yes

3. Has the statistical analysis been performed appropriately and rigorously? 

Reviewer #1: (No Response)

Reviewer #2: Yes

4. Have the authors made all data underlying the findings in their manuscript fully available?

Reviewer #1: (No Response)

Reviewer #2: Yes

5. Is the manuscript presented in an intelligible fashion and written in standard English?

Reviewer #1: (No Response)

Reviewer #2: Yes

6. Review Comments to the Author

Reviewer #1: (No Response)

Reviewer #2: (No Response)

7. PLOS authors have the option to publish the peer review history of their article (what does this mean?). If published, this will include your full peer review and any attached files.

Reviewer #1: **Yes: **João Cláudio Machado

Reviewer #2: No

---

## [Editor Report · Acceptance letter]

13 Nov 2020

PONE-D-20-14878R1 

Applications of a working framework for the measurement of representative learning design in Australian football 

Dear Dr. Browne:

I'm pleased to inform you that your manuscript has been deemed suitable for publication in PLOS ONE. Congratulations! Your manuscript is now with our production department. 

Kind regards, 

on behalf of

Dr. Slavko Rogan 

Academic Editor

PLOS ONE